# Soluble Receptor for Advanced Glycation End Products (sRAGE) Is a Sensitive Biomarker in Human Pulmonary Arterial Hypertension

**DOI:** 10.3390/ijms22168591

**Published:** 2021-08-10

**Authors:** Franziska Diekmann, Philippe Chouvarine, Hannes Sallmon, Louisa Meyer-Kobbe, Moritz Kieslich, Brian D. Plouffe, Shashi K. Murthy, Ralf Lichtinghagen, Ekaterina Legchenko, Georg Hansmann

**Affiliations:** 1Department of Pediatric Cardiology and Critical Care, Hannover Medical School, 30625 Hannover, Germany; Diekmann.Franziska@mh-hannover.de (F.D.); Chouvarine.Philippe@mh-hannover.de (P.C.); Louisa.Meyer-kobbe@stud.mh-hannover.de (L.M.-K.); Legchenko.Ekaterina@mh-hannover.de (E.L.); 2Department of Pediatric Cardiology, Charité University Medical Center, 13353 Berlin, Germany; sallmon@dhzb.de (H.S.); moritz.kieslich@charite.de (M.K.); 3Department of Chemical Engineering, Northeastern University, Boston, MA 02115, USA; brian.plouffe@regiscollege.edu (B.D.P.); Shashi.k.murthy@flaskworks.com (S.K.M.); 4Department of STEM, Regis College, Weston, MA 02493, USA; 5Flaskworks, LLC, Boston, MA 02118, USA; 6Institute of Clinical Chemistry, Hannover Medical School, 30625 Hannover, Germany; Lichtinghagen.Ralf@mh-hannover.de

**Keywords:** soluble receptor for advanced glycation end products (sRAGE), pulmonary arterial hypertension, biomarker, vascular injury, inflammation, proliferation, RV hypertrophy

## Abstract

Pulmonary arterial hypertension (PAH) is a progressive condition with an unmet need for early diagnosis, better monitoring, and risk stratification. The receptor for advanced glycation end products (RAGE) is activated in response to hypoxia and vascular injury, and is associated with inflammation, cell proliferation and migration in PAH. For the adult cohort, we recruited 120 patients with PAH, 83 with idiopathic PAH (IPAH) and 37 with connective tissue disease-associated PAH (CTD-PAH), and 48 controls, and determined potential plasma biomarkers by enzyme-linked immunoassay. The established heart failure marker NTproBNP and IL-6 plasma levels were several-fold higher in both adult IPAH and CTD-PAH patients versus controls. Plasma soluble RAGE (sRAGE) was elevated in IPAH patients (3044 ± 215.2 pg/mL) and was even higher in CTD-PAH patients (3332 ± 321.6 pg/mL) versus controls (1766 ± 121.9 pg/mL; *p* < 0.01). All three markers were increased in WHO functional class II+III PAH versus controls (*p* < 0.001). Receiver-operating characteristic analysis revealed that sRAGE has diagnostic accuracy comparable to prognostic NTproBNP, and even outperforms NTproBNP in the distinction of PAH FC I from controls. Lung tissue RAGE expression was increased in IPAH versus controls (mRNA) and was located predominantly in the PA intima, media, and inflammatory cells in the perivascular space (immunohistochemistry). In the pediatric cohort, plasma sRAGE concentrations were higher than in adults, but were similar in PH (n = 10) and non-PH controls (n = 10). Taken together, in the largest adult sRAGE PAH study to date, we identify plasma sRAGE as a sensitive and accurate PAH biomarker with better performance than NTproBNP in the distinction of mild PAH from controls.

## 1. Introduction

Pulmonary arterial hypertension (PAH) is a fatal disease, characterized by increased pulmonary vascular resistance (PVR) due to endothelial dysfunction, pulmonary vascular remodeling and vessel loss [1], leading to right ventricular dysfunction (RVD) [2]. Severe PAH results not only in RV hypertrophy (RVH) but also RV dilatation, and ultimately, RV failure [3,4]. The most common spontaneous or familial heterozygous loss-of-function mutations in heritable PAH (HPAH) occur in the bone morphogenetic protein receptor 2 (BMPR2) gene [5]. BMPR2 mutations are found in 70–80% of families with PAH and in 10–20% of idiopathic PAH patients [5]. Clinically, PAH manifests with non-specific symptoms, such as dyspnea and fatigue, making early diagnosis and the initiation of pharmacotherapy difficult, especially in children [6,7]. The late diagnosis, poor prognosis and complex etiology and pathophysiology of PAH underline the unmet need for sensitive biomarkers enabling early diagnosis, non-invasive monitoring, and risk stratification, in order to recognize disease progression for early intervention [8]. Numerous biomarkers of vascular injury and remodeling, myocardial damage, endothelial dysfunction and inflammation have been found to be associated with PAH [8,9,10,11]. So far, the N-terminal prohormone of the brain natriuretic peptide (NTproBNP) is the only prognostic biomarker included in international pulmonary hypertension (PH) treatment guidelines and risk scores [12,13]. However, NTproBNP does not distinguish between different PH etiologies and its levels are highly dependent on fluid intake and renal excretion [8]. The receptor for advanced glycation end products (RAGE) is a member of the immunoglobulin superfamily, and is a pattern recognition receptor (PRR) that binds damage- and stress-associated molecular patterns (DAMPs, “danger signals”) [14] released in response to hypoxia and vascular injury [15,16]. The binding of DAMPs to RAGE induces shedding of the membrane-bound RAGE into the circulation [16]. Increased soluble RAGE (sRAGE) concentrations indicate the overstimulation of RAGE by DAMPs, thereby reflecting the degree of ongoing inflammation and vascular damage [16,17]. Several animal and in vitro studies have identified RAGE to be associated with cellular key events in PAH development, i.e., inflammation, cell proliferation and migration [18,19,20,21]. Small exploratory clinical studies (n < 30 patients) found circulating sRAGE levels to be increased in idiopathic PAH (IPAH) vs. controls [19,22,23]. 

To the best of our knowledge, sRAGE concentrations circulating in the bloodstream have not yet been studied either in connective tissue disease-associated PAH (CTD-PAH) or in pediatric PH patients. Here, we report on the largest sRAGE biomarker study of human PAH (n = 120). We identify sRAGE as a sensitive biomarker in adult PAH that has comparable diagnostic accuracy to the established heart failure biomarker NTproBNP and shows even better performance in the distinction between mild PAH and controls. In contrast, plasma sRAGE was not specifically elevated in pediatric PH. 

## 2. Results

### 2.1. Demographic Characteristics of the Adult and Pediatric PAH Cohorts

We enrolled 120 adult patients (111 females, 9 males) with PAH and 48 healthy age-matched controls (29 females, 19 males) at the research conferences of the Pulmonary Hypertension Association (PHA) in California (2010), Florida (2012, 2018) and Texas (2016). Among the adult PAH patients, 83 had idiopathic pulmonary arterial hypertension (IPAH) (74 females, 9 males) and 37 connective tissue disease-associated PAH (CTD-PAH) (37 females, 0 males). The demographic characteristics of the adult female PAH patients and healthy age- and gender-matched controls under study can be found in Table 1. All characteristics of the male IPAH patients (n = 9) and male control subjects (n = 19), who were analyzed as small subcohorts separately, are presented in Appendix A. EDTA whole-blood samples were collected via peripheral venipuncture. For the pediatric cohort, we enrolled 10 children with PH (age range 3.9–18.5 years) and 10 non-PH, non-healthy control patients (9 with left ventricular outflow tract obstruction (LVOTO) and one with s/p reconstruction of a double aortic arch; age range 2.0–17.3 years) from October 2013 to August 2020. The children enrolled had moderate PH, with WHO functional class 2–3 and intermediate (n = 9) or lower risk (n = 1; Appendix A) according to the European Pediatric Pulmonary Vascular Disease Network (EPPVDN) risk score [13]. All 20 children underwent right and left heart catheterization. Detailed information on the pediatric PH-patients and non-PH controls can be found in Appendix A.

### 2.2. NTproBNP, IL-6 and sRAGE Plasma Levels Are Elevated in Adult IPAH and CTD-PAH Patients versus Healthy Control Subjects

In order to investigate whether NTproBNP, interleukin-6 (IL-6), and sRAGE plasma concentrations are elevated in adult patients with PAH versus healthy controls, we performed immunoassays on the plasma of IPAH patients, CTD-PAH patients and healthy controls. We focused on female patients in this study, but the biochemical results in the few male patients enrolled are presented in Appendix A. NTproBNP levels were markedly higher in both IPAH (316.8 ± 45.7 ng/L) and CTD-PAH patients (329.3 ± 65.0 ng/L) vs. controls (78.0 ± 11.5 ng/L; *p* < 0.001; Figure 1A,B). IL-6 concentrations were increased in patients with IPAH vs. controls (4.6 ± 0.5 ng/L vs. 2.8 ± 0.3 ng/L; *p* < 0.01) and CTD-PAH versus controls (4.5 ± 0.5 ng/L vs. 2.8 ± 0.3 ng/L; *p* < 0.05; Figure 1C,D). The plasma concentrations of sRAGE were elevated in IPAH patients (3044 ± 215.2 pg/mL; *p* < 0.01) and even higher in CTD-PAH patients (3332 ± 321.6 pg/mL; *p* < 0.001) as compared to controls (1766 ± 121.9 pg/mL; Figure 1E,F).

In the small cohort of male patients, we found elevated NTproBNP levels in IPAH patients (n = 9) vs. controls (n = 19; 284.3 ± 96.2 ng/L vs. 63.1 ± 26.3 ng/L; *p* < 0.01) while the elevation of IL-6 and sRAGE plasma concentrations in male IPAH patients versus male controls did not reach statistical significance (Appendix AA–F). Patients treated with prostacyclin or prostacyclin analogs (PCA) for more advanced PAH had 5-fold higher NTproBNP plasma concentrations (389.6 ± 62.0 ng/L vs. 78.0 ± 11.5 ng/L; *p* < 0.0001; Appendix A) and 2-fold higher sRAGE plasma levels versus controls (3481 ± 291.3 pg/mL vs. 1766 ± 121.9 pg/mL; *p* < 0.0001; Appendix A).

Subsequently, we evaluated whether plasma concentrations of NTproBNP, IL-6, and sRAGE increase with PAH severity, as defined by the World Health Organization (WHO) functional class system (FC I, n = 21; FC II+III, n = 90). NTproBNP, IL-6, and sRAGE plasma levels were significantly increased in FC II+III versus controls (*p* < 0.001 and *p* < 0.0001; Figure 2A–F). In addition, PAH FC II+III had higher IL-6 plasma concentrations than PAH FC I (*p* < 0.05; Figure 2C,D). We found a significant difference between PAH FC I and healthy controls for plasma sRAGE (2929 ± 421.8 vs. 1766 ± 121.9; *p* < 0.05) but neither for NTproBNP nor for IL-6 (Figure 2E,F). It is of note that circulating sRAGE concentrations in adult patients and control subjects are not age-dependent (Appendix A).

### 2.3. Receiver-Operating Characteristic (ROC) Analysis: Plasma sRAGE Has Diagnostic Accuracy Comparable to the Established Biomarker NTproBNP

To identify whether circulating sRAGE can serve as a biomarker in PAH, and to assess the diagnostic accuracy of sRAGE to classify subjects into different groups (PAH vs. CON, CTD-PAH vs. control, IPAH vs. control, FC I vs. control, FC II+III vs. control), we performed a receiver-operating characteristic (ROC) analysis (Figure 3). We compared the ROC for sRAGE with the ROC for the established heart failure biomarker NTproBNP and show the superimposed ROC curves for these two biomarkers in Figure 3. The fitted ROC area (AUC) in the PAH (IPAH + CTD-PAH) vs. control comparison was similar, at 0.767 for NTproBNP and 0.738 for sRAGE (Figure 3A). In the separate CTD-PAH vs. control and IPAH vs. control comparison, the fitted ROC area was only minimally higher for NTproBNP as compared to sRAGE (0.803 vs. 0.789 and 0.749 vs. 0.713; Figure 3B,C). In contrast, sRAGE outperformed NTproBNP in the PAH FC I (mild PAH) vs. healthy control comparison as the area under the ROC curve was 0.701 for sRAGE, compared to 0.677 for NTproBNP (Figure 3D). The fitted ROC area was slightly higher for NTproBNP in the PAH FC II+III vs. healthy control comparison, as compared to sRAGE (0.788 for NTproBNP vs. 0.747 for sRAGE; Figure 3E). Additionally, we performed a correlation analysis between plasma soluble RAGE and NTproBNP for all subjects, and found only a moderate correlation between these two biomarkers (rho = 0.4465, *p* < 0.0001; Appendix A).

### 2.4. RAGE mRNA and Protein Expression Is Increased in Lung Tissue from End-Stage IPAH Patients versus Donor Controls

Whole human lung tissues were obtained from 7 adult patients who underwent bilateral lung transplantation (LuTx) for end-stage PAH. Control lung tissues were obtained from 9 LuTx donors (downsizing lungs or unused donor lungs). Information on the LuTx subjects can be found in Table 2. 

To investigate the expression of RAGE in human lung tissues from PAH patients versus controls, RNA was extracted from human whole lung tissues, reverse transcribed into cDNA, and a real-time quantitative PCR was performed. The relative RAGE (*AGER*) mRNA expression was 2.1-fold higher in the whole lung tissues of end-stage IPAH patients (n = 7) versus controls (LuTx donors; n = 9; p < 0.05; Figure 4A). RAGE protein expression was elevated in the whole human lung tissues from end-stage IPAH patients (n = 5) vs. controls (n = 4), but this difference did not reach statistical significance (Figure 4B, Appendix A).

### 2.5. Immunohistochemistry RAGE Signal Is Augmented in the Pulmonary Vasculature of Adult IPAH Patients vs. Controls

The representative image of RAGE staining shows the boosted expression of RAGE in the intima, media and adventitia of obliterated distal pulmonary arteries (concentric hypertrophic lesion) and in perivascular cells in the outer adventitia of an end-stage IPAH patient (Figure 4F). The RAGE-protein-expressing (RAGE+) intravascular cells are probably mainly smooth muscle cells and fibroblasts, but also endothelial cells. The perivascular cells likely represent proinflammatory cells that are not only located in the interalveolar septa but are also infiltrating the adventitia of the concentric hypertrophic lesion.

### 2.6. Compartment-Specific Plasma Concentrations of NTproBNP, IL-6, and sRAGE in Children with PH versus Non-PH Controls in the Systemic and Pulmonary Circulation

In order to test whether NTproBNP, IL-6, and sRAGE levels are elevated in children with PH, immunoassays were performed in pediatric PH patients (n = 10) and non-PH controls (n = 10). To identify differences in circulating sRAGE levels across the hypertensive lung, sRAGE levels were determined in the superior vena cava (SVC), pulmonary artery (PA), and ascending aorta (AAO). The mild elevation of NTproBNP plasma levels in PH patients versus non-PH controls (SVC: 168.8 ± 50.8 ng/L vs. 79.6 ± 12.4 ng/L; PA: 172.7 ± 53.7 ng/L vs. 81.7 ± 12.9 ng/L; AAO: 170.3 ± 51.3 ng/L vs. 75.8 ± 14.5 ng/L) did not reach statistical significance (Figure 5A). We found a positive correlation of NTproBNP plasma concentrations (SVC) with a surrogate of disease severity, i.e., the ratio of mean pulmonary to systemic arterial pressure (mPAP/mSAP; rho = 0.57, *p* = 0.0085; Figure 5A), measured simultaneously with the blood draw. Except for four subjects, all IL-6 measurements in children were below the clinically reported detection range (PH vs. non-PH control: 2.5 ± 0.3 vs. 2.3 ± 0.2; data not shown). There was no statistically significant difference in the plasma compartment-specific concentrations of sRAGE in the SVC, PA, or AAO between PH-patients and non-PH controls (SVC: 5976 ± 1588 pg/mL vs. 4807 ± 653 pg/mL; Figure 5B). It is notable that sRAGE plasma levels in pediatric PH patients had a large variance (Figure 5B). We did not find a significant correlation of plasma sRAGE concentrations with mPAP/mSAP in children (Figure 5B). We also did not identify significantly different levels of sRAGE across the pulmonary circulation (AAO vs. PA) and the transpulmonary log2 fold changes of sRAGE did not correlate with the mean transpulmonary pressure gradient (mTPG; Figure 5C). The overall sRAGE levels were higher in children than in adults, both in non-PH controls and in children with moderate PH. To further analyze the role of aging in the context of sRAGE and PAH, we analyzed plasma sRAGE with respect to age in children, and found that circulating sRAGE levels tend to decrease with age (Appendix A). 

### 2.7. RAGE mRNA Expression in Explanted Lungs from Children with PAH and HPAH (Heterozygous BMPR2 Mutation)

Whole human lung tissues were obtained from 10 children who underwent bilateral LuTx for end-stage PAH (Appendix A). Quantitative analysis of RAGE (*AGER*) mRNA in children’s whole human lung tissues that were transplanted for either idiopathic PAH or pulmonary veno-occlusive disease (PVOD), vs. children with heritable PAH (HPAH, BMPR2 +/− mutation), revealed that the presence of a heterozygous BMPR2 gene mutation (HPAH, n = 4) did not influence the relative RAGE mRNA expression in HPAH-lungs vs. lungs from children that were transplanted for IPAH or PVOD/PAH (IPAH+PVOD n = 6; Figure 6A).

### 2.8. Strong Immunohistochemistry Signal for RAGE in the Intima and Media of Pulmonary Vessels of Children with PAH/PVOD or Heritable PAH (BMPR2 +/− Mutation) Undergoing Lung Transplantation

Although we did not find any differences in RAGE (*AGER*) transcripts between heritable and non-heritable pediatric PAH, we did find heightened RAGE expression in vascular and inflammatory cells in the intima, media and adventitia of obliterated pulmonary arteries of both children with IPAH and PVOD and those with HPAH-BMPR2 +/− (Figure 6B–E).

## 3. Discussion

An ideal biomarker is not only disease-specific, correlating with disease severity, progression and responsiveness to treatment, but is also non-invasive, reliable, valid and applicable in clinical practice [8]. Given the complex etiology and pathophysiology of pulmonary hypertension, a single biomarker (e.g., NTproBNP) can hardly be sufficient to fully reflect the patients’ stages of their disease, disease etiology, clinical prognoses, and risks of death. Recently, a multiple biomarker approach for PAH has been suggested [8,11,24], indicating that novel biomarkers besides NTproBNP need to be studied and validated in sufficiently sized cohorts. 

Here, we report on the largest sRAGE biomarker study in human PAH (n = 120). Plasma sRAGE, determined by enzyme-linked immunoassay, was 1.7-fold elevated in female IPAH patients versus controls. This study is the first that examines sRAGE plasma concentrations in adult patients with CTD-PAH, the second most common subtype in adult PAH [25,26]. We demonstrate 1.9-fold higher sRAGE levels in CTD-PAH compared to controls. sRAGE plasma concentrations in both adult IPAH and CTD-PAH correlated with the WHO functional class as a surrogate for disease severity. By applying ROC analysis for sRAGE versus the established heart failure biomarker NTproBNP, we show that sRAGE has comparable diagnostic accuracy and even better performance in the distinction between mild PAH (FC I) and healthy controls. These results point to the potential of sRAGE to serve as a useful biomarker in adult PAH, in addition to NTproBNP. In contrast, plasma sRAGE concentrations were similar between PH and non-PH, non-healthy controls, in a small number of children undergoing cardiac catheterization.

The hallmarks of PAH are inflammation, proliferation, migration, and subsequent vascular remodeling, as well as right ventricular hypertrophy and dilation due to increasing pressure afterload [1,2,3,4]. In all these pathobiological processes of pulmonary vascular disease (PVD), RAGE has been shown to be implicated, by means of preclinical studies and human tissue analysis [18,19,20,21]. As briefly stated above, RAGE binds damage- and stress-associated molecular patterns (DAMPs, “danger signals”) such as advanced glycation end products (AGEs), high-mobility group box 1 (HMGB1) and S100 proteins [14,16]. Previously, we discovered a hypoxia-driven, likely DAMP-induced miR-146b-TRAF6-IL-6/CCL2 (MCP-1) axis in the heart [27], by simulating PH-/RVH-associated coronary hypoxia in a murine alveolar hypoxia model. In the lung tissues of IPAH patients, resistin, the human homolog of RELMα (syn. HIMF, FIZZ1), is upregulated in macrophage-like inflammatory cells, in response to hypoxic damage and inflammation [21]. As a key DAMP, HMGB1 is released from endothelial cells (displaying dysfunction or apoptosis induced by HIMF) and thus promotes pulmonary artery smooth muscle cell (PASMC) proliferation in a RAGE-dependent manner [15]. RAGE exists as a membrane-bound full-length RAGE (FL-RAGE) and as soluble RAGE that is either produced by alternative splicing (esRAGE) or by shedding of the membrane-bound form, the so-called cleaved RAGE (cRAGE), which is the most common soluble form [17]. Increased soluble RAGE concentrations—as demonstrated here in adult IPAH and CTD-PAH patients’ blood plasma—are likely the result of RAGE shedding into the circulation following the overstimulation of RAGE by DAMPs [16,17]. Therefore, the increased circulating sRAGE concentrations in both IPAH and CTD-PAH patients as demonstrated in our current study are likely strongly associated with ongoing inflammation and vascular injury in adult PVD [16,17]. In a small clinical study on circulating sRAGE in both PAH (n = 14) and chronic thromboembolic pulmonary hypertension (CTEPH; n = 13), sRAGE plasma concentrations were increased compared to controls (*p* < 0.001), and sRAGE levels decreased after balloon pulmonary angioplasty in CTEPH patients (*p* < 0.001) [22]. In adult IPAH patients (n = 23), serum sRAGE levels were elevated and positively correlated with the mean pulmonary arterial pressure (mPAP) (r^2^ = 0.4542, *p* = 0.0004) [19], underlining the alterations and possible clinical importance of sRAGE in human PAH.

Besides circulating sRAGE, we demonstrate 2.1-fold higher RAGE mRNA expression and 1.9-fold higher protein expression in whole lung tissues from end-stage IPAH patients, when compared to control lungs. The magnitude of pulmonary sRAGE overexpression in our PAH study is consistent with a proteomic analysis of lung tissue homogenates from PAH patients (2.1-fold higher expression compared to controls) [28]. In addition, we located the RAGE protein expression signal predominantly in the intima and media of pulmonary arteries (SMC, myofibroblasts) and in inflammatory cells (lymphocytes, macrophages) in the adventitia and perivascular interstitial space. These results are consistent with the previously reported immunofluorescence findings that RAGE is expressed in PASMCs isolated from PAH patients’ lungs but is not expressed in PASMCs from patients without PAH [29].

In 2008, the nuclear hormone receptor and transcription factor, peroxisome proliferator-activated receptor gamma (PPARγ), was discovered as an anti-proliferative therapeutic target [30,31] that is activated by BMP2/BMPR2-signals in human PASMCs [31]. Subsequently, we identified PPARγ as the missing link between BMP2 and TGFβ1 pathways and unraveled a novel non-canonical TGFβ1-Stat3-FoxO1 axis in human PASMCs [32,33]. We could then demonstrate that the PPARγ agonist pioglitazone reverses PAH and prevents RV failure in SU5416/hypoxia (SuHx)-exposed rats [34]. Others showed that RAGE activation, triggered by S100A4, decreased BMPR2-PPARγ signaling in PASMCs from PAH patients through the activation of STAT3, and thus induced PASMC proliferation and resistance to apoptosis [18,35]. Moreover, the RAGE blockade in human PASMC suppressed the expression of pro-fibrotic extracellular matrix (ECM) proteins (collagen 1, tenascin-C, fibronectin) via the downregulation of TGFβ1 [19]. In both monocrotaline (MCT)-injected and SU5416/hypoxia-exposed rats, RAGE inhibition decreased PASMC proliferation and pulmonary artery medial thickness, as well as the mean pulmonary arterial pressure (mPAP) and RV hypertrophy [18]. A model of interdependent S100A4/Mts1-RAGE and BMP2/BMPR2 signaling has been proposed for human PASMC migration [36]. Exaggerated RAGE signaling leads to ERK phosphorylation and the induction of matrix metalloproteinase-2 (MMP2), and presumably the pathological migration of PASMCs in PVD [36]. Together with inositol monophosphatase 1 (IMPA1) as an interacting partner, RAGE appears to be actively involved in the vascular injury, cell proliferation and glycolytic shift that are characteristic of PAH [20]. Since the reported reduction of RV mass with RAGE-inhibition was also associated with decreased RV pressure afterload [18], it is unclear whether the RAGE-blockade can have a direct effect on RV mass, volume, and function. 

The reason for the similar sRAGE plasma concentrations in pediatric PH patients versus non-PH controls in our study may be multifactorial—from the small number of children enrolled, the non-healthy controls used as comparators, to further factors influencing the levels of circulating sRAGE, such as age, nutrition, medication, years since diagnosis/disease state, and the degree of inflammation. Moreover, the children had IL-6 levels below the clinically reported detection range. Given that RAGE is implicated in many proinflammatory signaling pathways [21], the degree of inflammation, as judged by circulating proinflammatory markers, appears to be less pronounced in children with mild to moderate PAH than in adults with IPAH or CTD-PAH. Overall, plasma sRAGE levels were higher in children than in adults, both in children with left ventricular outflow tract obstruction (LVOTO; non-PH controls) and in children with moderate PH. There are multiple possible explanations for this difference. In adults, EDTA blood was collected via peripheral venipuncture, whereas EDTA blood in children was collected during cardiac catheterization in the SVC, PA and AAO. Given that RAGE is highly expressed in the lungs [37], the reason for the higher sRAGE levels in children might possibly be the anatomically closer blood draw next to the lungs (pulmonary artery, aorta). However, the fact that the pediatric SVC sRAGE levels were also more than 2-fold higher in children than the peripherally venous sRAGE in adults argues against this speculation. Importantly, in the adult cohort, healthy adults served as control. In the pediatric cohort, however, patients with repaired/residual congenital heart disease (LVOTO or s/p double aortic arch) served as non-PH controls. Although we did show that aging in adults did not influence the sRAGE plasma levels, it is possible that sRAGE concentrations are generally higher in children compared to adults, indicating that sRAGE is developmentally regulated. Others found that sRAGE levels, measured in the bronchoalveolar lavage fluid (BALF) of children but not in plasma, were inversely correlated to age [38]. We also found a trend toward lower sRAGE plasma levels regarding aging in children, getting closer to the sRAGE levels measured in adults.

Our study has several limitations, most of which are inherent to biomarker studies in a rare disease, including the lack of invasive hemodynamics in adult PAH patients (n = 120) and the small number of PH patients in the pediatric cohort (n = 10). In addition, the lack of lung tissues and invasive hemodynamics from healthy children make it difficult to compare the sRAGE plasma and tissue expression levels of pediatric PH patients with controls. Larger prospective studies are needed to investigate the role of circulating sRAGE and cardiopulmonary RAGE expression and signaling in children.

Taken together, we report the largest sRAGE biomarker study in human adult PAH (n = 120), and the first determination of plasma sRAGE in CTD-PAH. We identify sRAGE as a sensitive biomarker in adult PAH, with comparable diagnostic accuracy to the established heart failure biomarker NTproBNP, and even better performance in the distinction between mild PAH and controls. Therefore, we suggest circulating sRAGE as an additional biomarker for use in clinical practice to diagnose and monitor adult PAH for response to therapy, disease progression, and early intervention.

## 4. Materials and Methods

### 4.1. Clinical Study Design

Adult PAH cohort: controls and PAH disease groups were well matched in terms of age, gender and BMI. The BMI varied only mildly, between 26.7 (IPAH), 27.7 (CTD-PAH) and 27.8 (controls; Table 1). A BMI > 25 in the United States classifies both the controls and PAH patients on average to be overweight (BMI 25.0–29.9). First-degree relatives and adult patients with sleep apnea, liver disease, chronic obstructive pulmonary disease (COPD), pulmonary fibrosis and congenital heart disease were excluded. We excluded one female patient due to the contradictory result of an NTproBNP level of 2565 ng/L and subjective WHO Functional Class I. A questionnaire concerning demographics and medications was completed by each subject. The WHO FC classification was a patient self-assessment during the patient interview at the PHA research conferences. 

Pediatric cohort: during cardiac catheterization, EDTA blood was collected near-simultaneously at three anatomic blood draw sites, together with pressure recordings and blood gas analysis (SpO2): superior vena cava (SVC), pulmonary artery (PA) and ascending aorta (AAO), as previously described [39,40]. Patients (PH, non-PH controls) with any intra- or extracardiac shunt were excluded. Written informed consent was obtained from the legal caregivers of each study subject. Pulmonary hypertension was defined according to the recent World Symposium on Pulmonary Hypertension (WSPH) in Nice (2018): mPAP > 20 mmHg [41,42].

### 4.2. Biomarker Assays

After the blood draw, EDTA whole blood samples were immediately processed for plasma by centrifugation at 1300× *g* for 10 min at room temperature. Plasma was then aliquoted and stored at −80 °C until use. N-terminal pro B-type natriuretic peptide (NTproBNP) and Interleukin-6 (IL-6) levels were measured using the Cobas e 801 immunoassay analyzer (Roche Diagnostics, Mannheim, Germany, NTproBNP: #07027664190; IL-6: #07027532190) that is in routine clinical use. The soluble receptor for advanced glycation end products (sRAGE) levels were determined in adult subjects (no dilution) and pediatric subjects (1:4 dilution) using the Human RAGE Immunoassay (ELISA) (#DRG00, R&D Systems, Minneapolis, MN, USA) according to the manufacturer’s instructions. Standard curves were determined using the nonlinear least squares regression analysis (the nls function) in R and the sample concentration values were calculated based on the corresponding standard curves. In the adult cohort, 19 measurements were above the detection range of the standard curve, so concentrations were determined by nonlinear least squares regression analysis. A detailed description of methods is provided as Appendix A.

### 4.3. RNA Extraction and Quantitative Real-Time PCR

Around 50–75 mg of whole human lung tissues were pre-treated with RNAlater-ICE Solution (Invitrogen, Life Technologies, Carlsbad, CA, USA, AM7030) and processed with a Polytron tissue homogenizer. RNA was then extracted according to the TRIzol protocol (TRIzol, Life Technologies, Carlsbad, CA, USA). Using the Nanodrop 2000c (Thermo Scientific), RNA concentrations were determined spectrophotometrically, and all samples underwent RNA quality control (RIN > 6). To generate the first-strand cDNA, the SuperScript III First-Strand Synthesis SuperMix for qRT-PCR was used according to the manufacturer’s instructions (Invitrogen by Life Technologies, Carlsbad, CA, USA, #11752). The quantitative PCR was run in triplicates using the TaqMan Universal Master Mix II (Applied Biosystems, Thermo Scientific, Vilnius, Lithuania, #4440040). TaqMan primers for AGER (Life Technologies, Carlsbad, CA, USA, Cat#Hs00542584_g1) and GAPDH (housekeeper; Life Technologies, Carlsbad, CA, USA, Cat#Hs02758991_g1) were used.

### 4.4. Protein Extraction and Western Blot

100–150 mg of whole human lung tissue was added to lysis buffer (Complete Lysis-M, EDTA-free, Roche Diagnostics, Mannheim, Germany, #04719964001), supplemented with anti-phosphatase and anti-protease inhibitors (PhosphoSTOP EASYpack, Roche Diagnostics, Mannheim, Germany, #04906845001). The tissues were homogenized in lysis buffer and hemolysis tubes were spun down for 3 min at 4000× *g* rpm (4 °C). The lysate was collected in 1.5 mL tubes and then centrifuged at 12,000× *g* for 10 min. The supernatant was stored at −80 °C until use. Protein concentrations were determined using the Pierce BCA Protein Assay Kit (Thermo Scientific, Rockford, IL, USA, #23225). For analysis, 40 µg of proteins were loaded into each lane of a NuPAGE 4–12% Bis-Tris Gel (Invitrogen by Thermo Fisher Scientific, Life Technologies Corporation, Carlsbad, CA, USA, #NP0335BOX). Gel electrophoresis was performed under reducing conditions. The membrane was blocked for one hour at room temperature with 5% non-fat milk in TBS containing 0.1% Tween, followed by incubating with 2 µg/mL of mouse anti-human RAGE monoclonal antibody (R&D Systems, Minneapolis, MN, USA, Cat #MAB1145) at 4 °C overnight in 5% non-fat milk in TBS-Tween. The next day, the membrane was washed with TBS-Tween and incubated with secondary HRP-linked anti-mouse antibody (Cell Signaling, Danvers, MA, USA, #7076, dilution 1:1000) at room temperature for one hour. SuperSignal West Pico PLUS Chemiluminescent Substrate (Thermo Scientific, Rockford, IL, USA, #34577) was used to visualize the binding of the secondary HRP-antibody. The blot was then stripped with Restore PLUS Western Blot Stripping Buffer (Thermo Scientific, Rockford, IL, USA, #46430), washed with TBS-T and incubated with GAPDH as a loading control (Santa Cruz Biotechnology, Dallas, TX, USA, sc-25778, dilution 1:1000) at 4 °C overnight. The next day, the membrane was washed with TBS-Tween and incubated with secondary HRP-linked anti-rabbit antibody (Cell Signaling, Danvers, MA, USA, #7074, dilution 1:1000). The Western blots were analyzed with ImageJ (Fiji), and the optical density values were normalized to GAPDH.

### 4.5. Immunohistochemistry

For immunohistochemistry, human lung sections were obtained from IPAH patients, HPAH patients and controls (LuTx donor lungs). The lung sections were deparaffinized and rehydrated. Antigen retrieval was performed by boiling the slides in antigen retrieval buffer (ab93678, abcam) at 96 °C in a water bath for 20 min. Peroxidase was blocked by incubating the slides with 3% hydrogen peroxide in H_2_O for 20 min. The slides were then washed with PBS containing 0.05% Tween 20 (PBS-T) and blocked for one hour with PBS-T containing 5% Normal Donkey Serum. Tissues were incubated with primary antibody against RAGE (Santa Cruz Biotechnology, Dallas, TX, USA, sc-365154) diluted in PBS-T containing 5% Normal Donkey Serum (dilution 1:50) at 4 °C overnight. The next day, the slides were washed with PBS-T (0.05% Tween) and incubated with secondary antibody (Santa Cruz Biotechnology, Dallas, TX, USA, sc-516102) diluted in PBS-T containing 5% Normal Donkey Serum (dilution 1:50) for one hour at room temperature. The slides were again washed with PBS-T (0.05% Tween) and 3′,3′-diaminobenzidine (DAB) was used as HRP-sensitive substrate solution. The lung sections were counterstained with hematoxylin and were finally dehydrated before mounting with DPX.

### 4.6. Statistical Analysis

The statistical analysis was performed in the GraphPad Prism software (Version 6.0) and in R. For transpulmonary gradient analysis, we used the mixed-effects models with log2 of fold change (FC) between two catheterization sites (AAO vs. PA) as the dependent variable, the groups (PAH or Control) as an independent variable, and each patient as a random effect (log2(FC)~Group, random = ~1|Patient). Given a relatively small sample size, we set the parameter sigma to 10, to remove the most improbable values. The generated models were evaluated using the Anova function from the car R package, and the *p*-values generated by the Wald chi-square test (using the car R package). Normal distribution was tested with D’Agostino and Pearson omnibus, Shapiro–Wilk, and Kolmogorov–Smirnov normality tests. For two-group comparisons, we used a two-tailed Welch’s *t*-test if the data passed all three normality tests, or the Mann–Whitney U test otherwise. The three-group comparisons were performed with the Kruskal–Wallis test, corrected for multiple testing by Dunn’s test. Data are presented as the mean ± standard error of the mean (SEM) or as the median with interquartile range (IQR). *p* < 0.05 was considered significant. Receiver operating characteristic (ROC) graphs were created using the plotROC R package. The corresponding area under the curve (AUC) was calculated using the pROC R package. For correlation analysis, the normal distribution of the two variables was tested by performing the Shapiro–Wilk test, using the mshapiro.test function from the mvnormtest R library. Depending on the outcome of this normality test, either the Pearson or Spearman correlation test was performed.

## Figures and Tables

**Figure 1 ijms-22-08591-f001:**
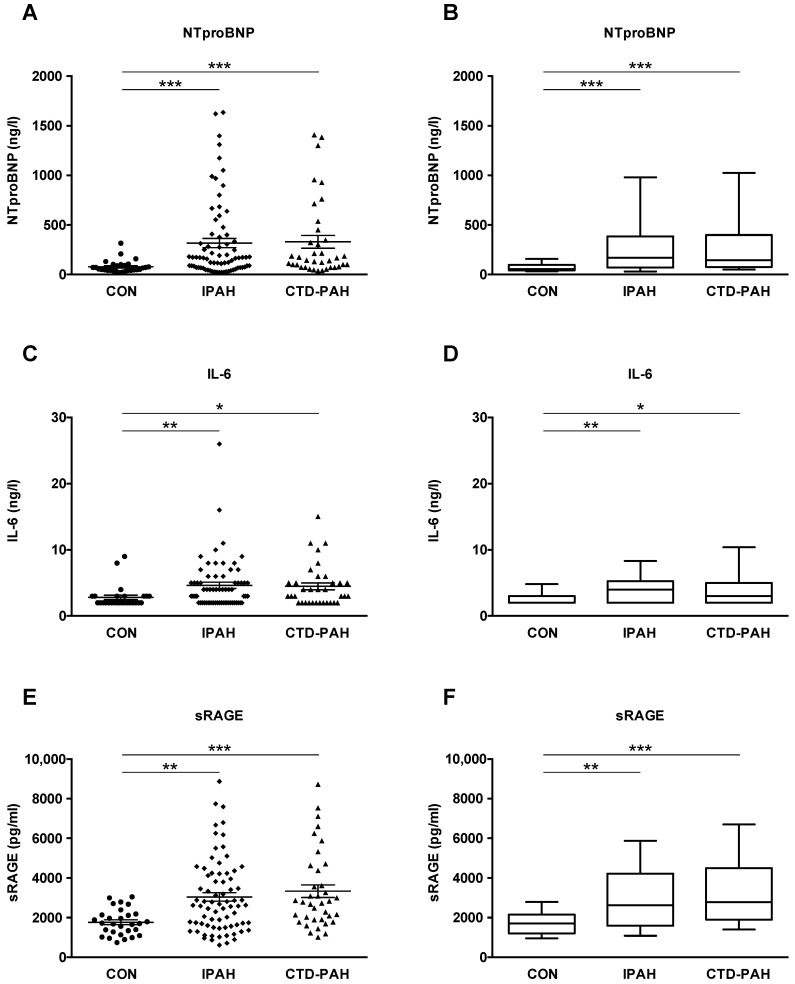
NTproBNP, IL-6 and sRAGE plasma concentrations are elevated in patients with pulmonary arterial hypertension versus controls. (**A**,**B**) NTproBNP plasma concentrations of patients with idiopathic pulmonary arterial hypertension (IPAH, n = 74) and connective tissue disease-associated pulmonary arterial hypertension (CTD-PAH, n = 37) versus controls (CON, n = 29). (**C**,**D**) IL-6 plasma concentrations of IPAH (n = 66) and CTD-PAH (n = 35) patients versus controls (n = 27). (**E**,**F**) sRAGE plasma concentrations of IPAH (n = 74) and CTD-PAH (n = 37) patients versus controls (n = 29). The scatter plots on the left show the mean ± SEM, the box and whisker plots on the right show the median with interquartile range ± 10–90 percentile. Statistical test: Kruskal–Wallis test, corrected for multiple testing by Dunn’s test. * *p* < 0.05, ** *p* < 0.01, *** *p* < 0.001. Abbreviations: IL-6, Interleukin-6; NTproBNP, N-terminal pro-brain natriuretic peptide; sRAGE, soluble receptor for advanced glycation end products.

**Figure 2 ijms-22-08591-f002:**
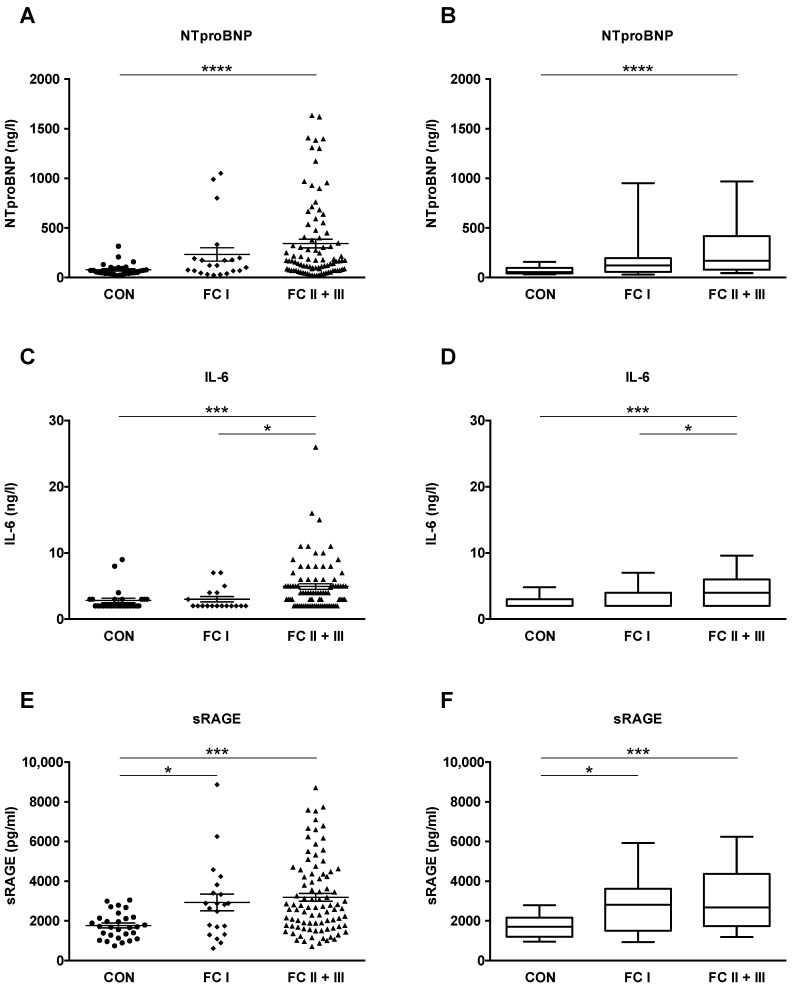
NTproBNP, IL-6 and sRAGE plasma concentrations in patients with idiopathic pulmonary arterial hypertension (IPAH) and connective tissue disease-associated pulmonary arterial hypertension (CTD-PAH) increase with pulmonary hypertension severity. The World Health Organization (WHO) functional class (FC) system is used as a surrogate for disease severity. Functional classes I-III include both IPAH and CTD-PAH patients. (**A**,**B**) Plasma levels of NTproBNP in controls (n = 29), patients with FC I (n = 21) and FC II and III (n = 90). (**C**,**D**) Plasma levels of IL-6 in controls (n = 27), patients with FC I (n = 18) and FC II and III (n = 83). (**E**,**F**) Plasma levels of sRAGE in controls (n = 29), patients with FC I (n = 21) and FC II and III (n = 90). The scatter plots on the left show the mean ± SEM, the box and whisker plots on the right show the median with interquartile range ± 10-90 percentile. Statistical test: Kruskal–Wallis test, corrected for multiple testing by Dunn’s test. * *p* < 0.05, *** *p* < 0.001. **** *p* < 0.0001. Abbreviations: IL-6, Interleukin-6; NTproBNP, N-terminal pro-brain natriuretic peptide; sRAGE, soluble receptor for advanced glycation end products.

**Figure 3 ijms-22-08591-f003:**
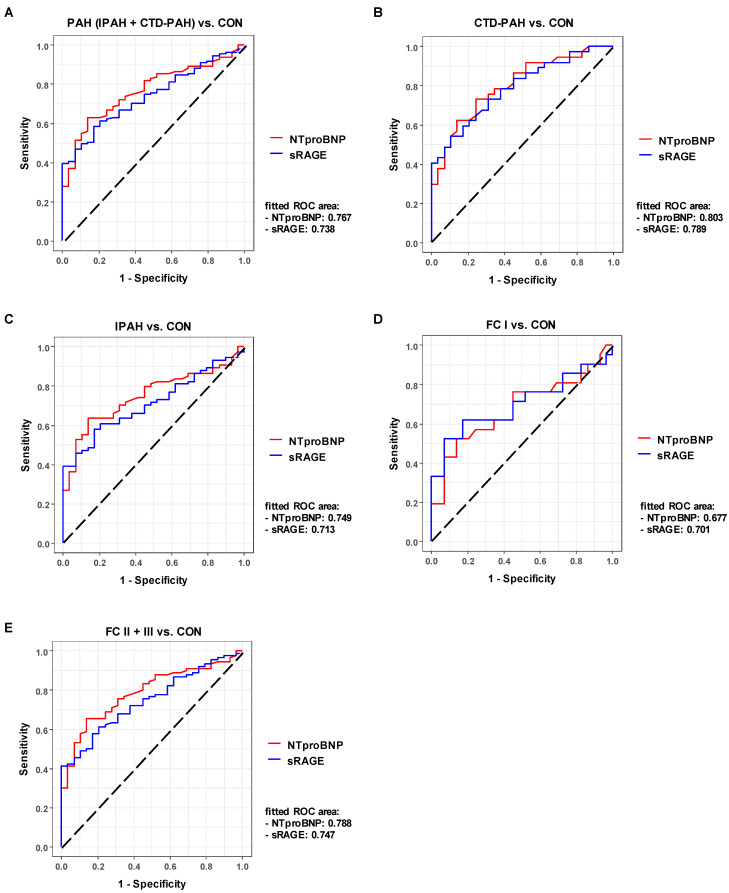
Plasma sRAGE has diagnostic accuracy in adult PAH comparable to the gold standard NTproBNP. Comparisons of various PAH groups vs. control (**A**–**C**) and functional classes vs. control (**D**,**E**) illustrate that sRAGE has comparable diagnostic accuracy to NTproBNP and even outperforms NTproBNP in the FC I vs. control comparison (**D**), i.e., AUC_sRAGE_ > AUC_NTproBNP_. Abbreviations: AUC, area under the ROC curve; NTproBNP, N-terminal pro-brain natriuretic peptide; ROC, receiver operating characteristic; sRAGE, soluble receptor for advanced glycation end products.

**Figure 4 ijms-22-08591-f004:**
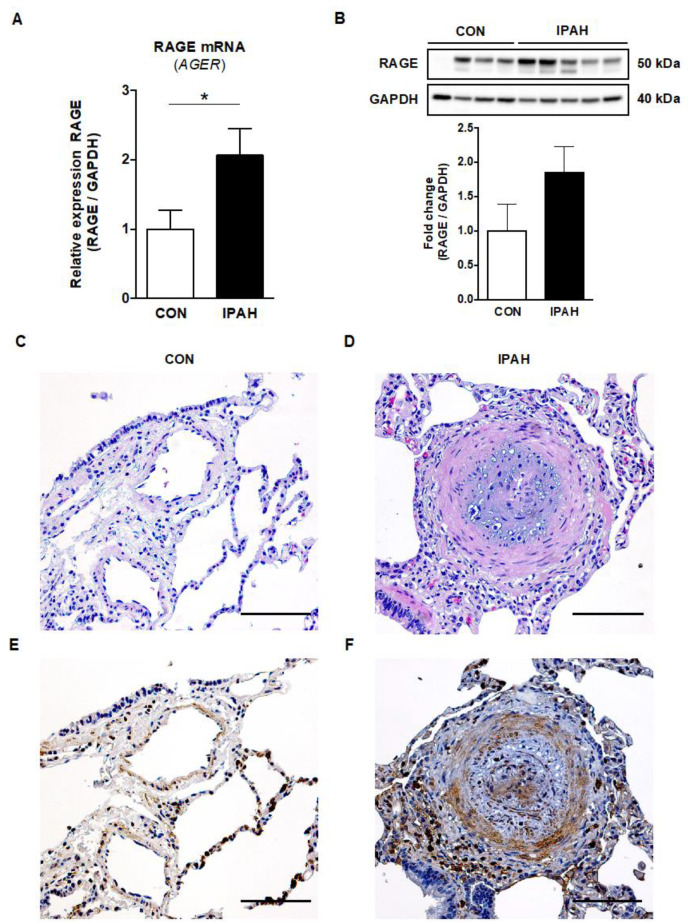
RAGE mRNA and protein expression, and immunohistochemistry in human lung tissues from end-stage IPAH patients versus controls. (**A**) The relative RAGE mRNA expression (AGER) normalized to GAPDH was significantly increased in end-stage IPAH patients (n = 7) vs. controls (CON; n = 9). (**B**) RAGE protein expression was elevated in whole human lung tissues of end-stage IPAH patients (n = 5) vs. controls (n = 4) measured by Western blot, but this difference did not reach statistical significance. Exposure time: 3.0 s. Values are presented as mean ± SEM. Statistical test: unpaired *t*-test with Welch’s correction, Mann–Whitney U. * *p* < 0.05. (**C**) Pulmonary artery of a female lung transplant donor lung (H&E staining; scale bar, 100 µm). (**D**) Obliterated pulmonary artery of an end-stage 42-year-old female IPAH patient (H&E staining; scale bar, 100 µm). (**E**) RAGE staining in a lung transplant donor lung of a female donor (scale bar, 100 µm). (**F**) The representative image of RAGE staining in a 42-year-old female IPAH patient shows the boosted RAGE expression in the intima and media (endothelial cells, smooth muscle cells, fibroblasts) of an obliterated distal pulmonary artery (concentric hypertrophic lesion). Heightened RAGE expression is also evident in perivascular cells in the outer adventitia, likely representing infiltrating proinflammatory cells, such as macrophages and lymphocytes (scale bar, 100 µm).

**Figure 5 ijms-22-08591-f005:**
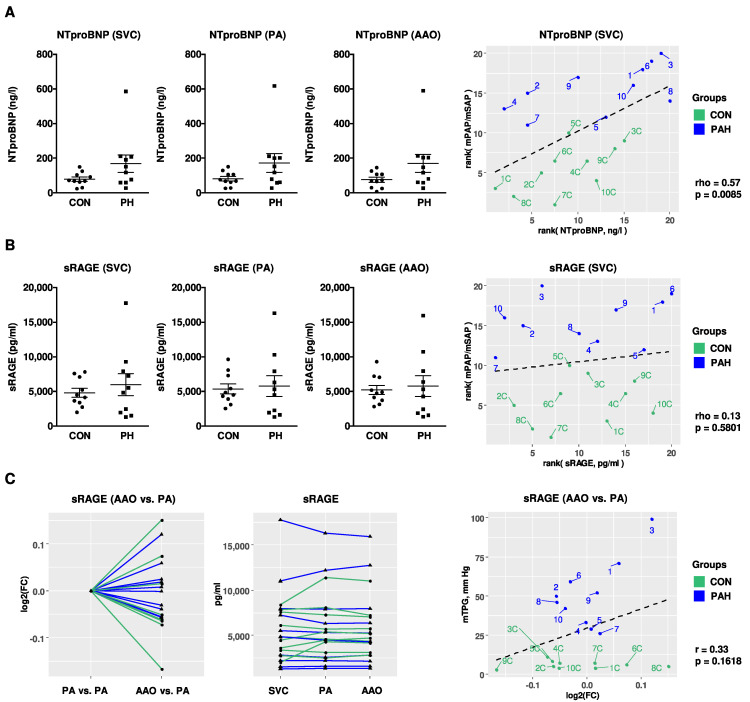
Compartment-specific blood plasma concentrations of NTproBNP and sRAGE in children with PH vs. non-PH controls in the systemic and pulmonary circulation. (**A**,**B**) NTproBNP and sRAGE plasma concentrations in the SVC, PA and AAO of pediatric patients with pulmonary hypertension (PH; n = 10) versus non-PH controls (CON; n = 10) and correlations with mPAP/mSAP. For measurement of sRAGE, samples were diluted 1:4, followed by enzyme-linked immunoassay (ELISA). (**C**) There were no different levels of sRAGE across the pulmonary circulation (AAO vs. PA) and the transpulmonary log2 fold changes of sRAGE do not correlate with mTPG. Data are shown as mean ± SEM. Statistical test: Mann–Whitney U test. Abbreviations: AAO, ascending aorta; CON, control; HPAH; heritable pulmonary arterial hypertension; IPAH, idiopathic pulmonary arterial hypertension; mPAP, mean pulmonary artery pressure; mSAP, mean systemic arterial pressure; mTPG, mean transpulmonary pressure gradient; PA, pulmonary artery; SVC, superior vena cava.

**Figure 6 ijms-22-08591-f006:**
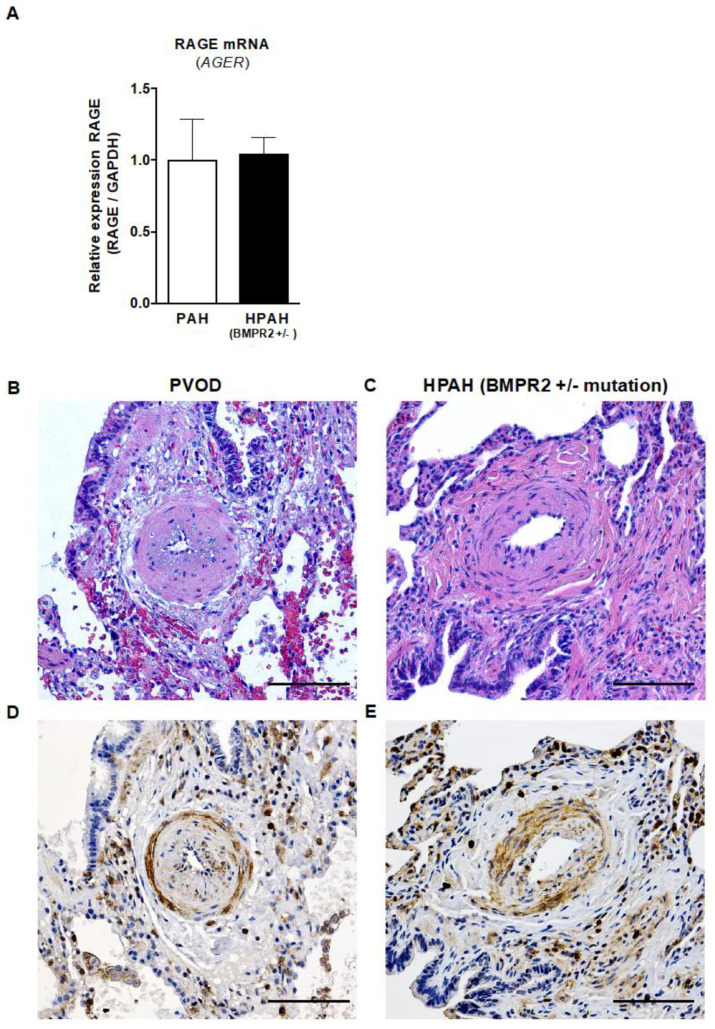
RAGE mRNA expression and immunohistochemistry in children with end-stage PAH and HPAH (heterozygous BMPR2 mutation). (**A**) The relative RAGE mRNA expression in children transplanted for idiopathic PAH or pulmonary veno-occlusive disease (IPAH + PVOD = PAH, n = 6) vs. children with heritable PAH (HPAH, BMPR2 +/− mutation, n = 4) shows that a heterozygous BMPR2 mutation does not influence the relative RAGE mRNA expression in whole lung tissues from patients undergoing lung transplantation. Values are presented as mean ± SEM. Statistical test: Mann–Whitney U. (**B**,**C**) H&E images of distal pulmonary arteries of an 11-year-old patient with PVOD (B) and a 5-year-old patient with BMPR2 +/− mutation (HPAH; C; scale bar, 100 µm). (**D**,**E**). The corresponding images of RAGE staining show increased RAGE expression in vascular and inflammatory cells in the intima, media and adventitia of pulmonary arteries in both the 11-year-old patient with PVOD (D) and the 5-year-old patient with BMPR2 +/− mutation (E; scale bar, 100 µm). Abbreviations: BMPR2, bone morphogenetic protein receptor 2; PAH, pulmonary arterial hypertension.

**Table 1 ijms-22-08591-t001:** Characteristics of PAH patients and healthy controls.

	CON (n = 29)	PAH Total (n = 111)	IPAH (n = 74)	CTD-PAH (n = 37)
**Demographics**				
Age—years	44.8 (21–76)	48.7 (20–80)	47.6 (20–79)	50.0 (26–80)
Male sex—n	0	0	0	0
Height—m	1.64 ± 0.01	1.63 ± 0.01	1.63 ± 0.01	1.63 ± 0.01
Weight—kg	74.7 ± 3.6	71.9 ± 1.5	71.0 ± 2.0	73.8 ± 2.2
BMI—kg/m^2^	27.8 ± 1.4	27.0 ± 0.5	26.7 ± 0.7	27.7 ± 0.8
**Functional Status**				
WHO FC I—n (%)	-	21 (19%)	20 (27%)	1 (3%)
WHO FC II—n (%)	-	67 (60%)	40 (54%)	27 (73%)
WHO FC III—n (%)	-	23 (21%)	14 (19%)	9 (24%)
**Biomarker**				
NTproBNP—ng/L	78.0 ± 11.5	321.0 ± 37.2	316.8 ± 45.7	329.3 ± 65.0
**Race/ethnicity**				
White	21	80	56	24
Black	3	7	2	5
Asian	1	6	3	3
Hispanic	3	8	6	2
other	1	10	7	3

Values are presented as the number of subjects or as mean ± SEM. Abbreviations: BMI, body mass index; CTD-PAH, PAH associated with connective tissue disease; IPAH, idiopathic pulmonary arterial hypertension; NTproBNP, N-terminal prohormone of brain natriuretic peptide; WHO FC, World Health Organization Functional Class.

**Table 2 ijms-22-08591-t002:** Human lung tissues from adult end-stage PAH patients and controls.

Group	Gender	Age (Years)	Diagnosis
IPAH	female	41	IPAH
IPAH	female	31	IPAH
IPAH	female	29	IPAH
IPAH	female	42	IPAH
IPAH	female	36	IPAH
IPAH	female	53	IPAH
IPAH	female	25	IPAH
Control	female	N/A	Downsizing lung
Control	male	N/A	Unused donor lung
Control	female	N/A	Unused donor lung
Control	male	N/A	Unused donor lung
Control	male	N/A	Downsizing lung
Control	N/A	N/A	Unused donor lung
Control	N/A	N/A	Unused donor lung
Control	male	N/A	Unused donor lung
Control	N/A	N/A	Downsizing Lung

Whole human lung tissues from patients who underwent bilateral lung transplantation for end-stage PAH and human lung tissues from donor lungs (downsizing lung or unused donor lung). Random peripheral lung tissue that was not close to the hilus, the main branch pulmonary arteries, and main bronchi was obtained from each lung. Abbreviations: IPAH, idiopathic pulmonary arterial hypertension; PAH, pulmonary arterial hypertension.

## Data Availability

There are no additional publicly achieved datasets.

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
