# Peer review of "Soluble Receptor for Advanced Glycation End Products (sRAGE) Is a Sensitive Biomarker in Human Pulmonary Arterial Hypertension"

_ijms, 2021, doi:10.3390/ijms22168591_

Round 1

Reviewer 1 Report

In this study, authors examined the relationship between the soluble receptor for advanced glycation end products (sRAGE) and pulmonary arterial hypertension (PAH) using 48 healthy subjects as controls and 120 PAH patients [83 idiopathic PAH (IPAH) and 37 connective tissue disease-associated PAH (CTD-PAH)]. They tested and compared the serum levels of N-terminal prohormone of brain natriuretic peptide (NTproBNP), IL-6, and sRAGE using PAH patients and controls. Among all, NTproBNP is currently used as a prognostic marker for PAH. Authors showed that the increase of serum sRAGE level was comparable to the increase of serum NTproBNP level found in PAH adult patients, as compared to the control group. They then examined adult lung tissues of end-stage IPAH, and further tested sRAGE using PAH children. The authors concluded that sRAGE can be used as a PAH biomarker and a better marker for the mild PAH. 

It is a constructive path to discover new biomarker(s) for PAH patients, and this is an interesting study. Although authors mentioned that there were 3 similar studies with smaller sample numbers (n<30), yet this study did use a larger cohort of adult PAH patients. However, there is missing information/data such as the control. Some results/conclusions should be clarified. In addition, this manuscript contains many errors and inconsistencies.  

Concerns:

  • In Fig 6, the authors determined RAGE mRNA level and the protein level using lung tissues from children with end-stage PAH (n=6) and HPAH (n=4). The authors need to establish/include the baseline data from health children as the control for comparison.

  • In Figs 6 and 4, are the detected RAGE signals (mRNA and protein levels) mainly coming from pulmonary arteries or inflammatory cells, or equally from both? The RAGE can locate in pulmonary arteries as well as the lymphocytes and macrophages in the adventitia and perivascular interstitial space. Additionally, in lines 218-219, the author “speculated” RAGE signal likely from endothelial cells, smooth muscles, and fibroblasts.

Therefore, this question should be examined, and the results can provide the clarification. Furthermore, this data may also provide insight about the huge differences of sRAGE baselines between adult and child healthy subjects (see the next question).

  • In Fig 5, the serum level of NTproBNP in healthy children is similar to that of healthy adults (about 80 ng/L), whereas the sRAGE serum level in healthy children (around 5000 ng/L) is about 2.5-fold higher than that of healthy adults (about 1800 ng/L). Why did this discrepancy occur?

  • In addition, in Fig 5 only the plot of NTproBNP serum level vs mPAP/mSAP (mean pulmonary arterial pressure vs mean systemic arterial pressure) from superior vena cava area (SVC) shows the significant difference (p=0.0085) between PAH children vs control, but not using sRAGE data. Therefore, these results did not support sRAGE as a good biomarker, especially for PAH children. The NTproBNP is still a better marker comparatively. The authors should re-evaluate their statements and conclusion in the manuscript.

  • There are several inconsistencies between the figs and fig legends, as well as statements. The missing information/narrations in fig legends and corresponding texts are also found.

Just name a few here:

  • In Fig 5, there is no IL6 data included, but the figure legend mentioning IL-6.
  • Are Fig 2A, C, E and Fig 2B, D, F related to IPAH and CTD-PAH, respectively? Or vice versa? The statement should be provided in the fig legend and the responding text.
  • In Fig 4 legend, there is no narration of Fig 4E. Also, the RAGE staining (D) is wrongly labeled. It should be E and F (line 231).
  • Fig 4B, is the statistically significant? There is no asterisk labeled. The corresponding text should be checked.
  • In Fig 6 fig legend, the narrations of A-E are not in ordered.
  • As mentioned in the concern #2, the author “speculated” the RAGE signals likely from endothelial cells, smooth muscles, and fibroblasts (in the Results section). However, in the Discussion section (lines 348-349) the authors stated, “we located the RACE protein expression … predominantly in the intima and media of pulmonary arteries and ….”.  These are conflicted statements.  Without the further empirical data support, how can the authors confirm the cell types expressing RAGE?
  • The authors mentioned the heterozygous loss-of-function BMPR2 mutation can found PAH patients. Is this commonly found in PAH patients? What is the percentage of PAH patients possessing BMPR2+/- mutation?
  • What are the criteria used to classify WHO function class I, II and III? This information should be briefly narrated in the manuscript.

Finally, there are more errors/typos, which are not mentioned. The authors should carefully check their manuscript.  

Reviewer 2 Report

This MS by Diekmann et al brings whole lot of new information to the PAH field, detailing the dynamic role of soluble receptor for advanced glycation end products (sRAGE) as a biomarker for this disease. Authors have included mostly female patients in their investigation. Authors have also assessed the role of sRAGE as a biomarker in a variety of PAH subtype including iPAH, HPAH and CTD-PAH, and in details according to the functional class, which is the biggest merit of this findings. Authors have also assessed the contribution of other known biomarkers of PAH (NTproBNP, IL-6) along with sRAGE and have done ROC analysis. To confirm the signals at local tissue, authors have performed molecular characterization using RT-PCR and western blots. Here are some minor comments -

  1. What is the rationale behind including a predominant number of female patients in the study? Can authors increase the n for male patients?
  2. What is the rationale for discussing combined results of FC II and III? Authors are encouraged to discuss the same, as a stand-alone group
  3. Did authors investigate other metabolic profile or characteristics of this patients? How many of the patients are diabetes or obese? What is the BMI of these patients? What is the glucose level of the entire cohort? Answers to the above questions may shed more light into the current outcome!
  4. Authors may bring in more information in the discussion to talk about the role of aging in the context of sRAGE and PAH.
  5. In addition to ROC, did authors assess a correlation between NTproBNP vs sRAGE or IL-6 vs sRAGE? A graphical representation of such outcome will be a value addition to the existing dataset in the Figure 3.
  6. Which portion of the lung chunk was chosen for molecular characterization or morphometric staining? Distal or region closer to main branch of the pulmonary artery or any random region? These details are highly critical and please disclose the same in the methods. 

Reviewer 3 Report

This manuscripts examines the correlation between soluble receptor of advanced glycation end products (sRAGE) and pulmonary arterial hypertension (PAH) across human samples. Overall the manuscript provides convincing data and examined the degree of PAH and how they relate to sRAGE levels. The manuscript is well written with mostly just issues stemming from the abstract which requires further attention and is poorly written. 

Line 34: Change by to using immunohistochemistry, located

Line 35: Change to: intima, media and inflammatory

Line 37: Change to “In the largest adult sRAGE study to date”

Abstract: Denote soluble RAGE prior to using sRAGE

Line 176: no need to spell out LuTx twice.
